Turtles of the genera Geoemyda and Pangshura (Testudines: Geoemydidae) lack differentiated sex chromosomes: the end of a 40-year error cascade for Pangshura

Mazzoleni Sofia 1
Augstenová Barbora 1
Clemente Lorenzo 1
Auer Markus 2
http://orcid.org/0000-0002-6740-7214 Fritz Uwe 2
Praschag Peter 3
Protiva Tomáš 4
Velenský Petr 5
http://orcid.org/0000-0002-3515-729X Kratochvíl Lukáš 1
http://orcid.org/0000-0002-8429-5680 Rovatsos Michail 1 michail.rovatsos@natur.cuni.cz
1 Department of Ecology, Charles University , Prague , Czech Republic
2 Museum of Zoology, Senckenberg , Dresden , Germany
3 Turtle Island , Graz , Austria
4 landsnails.org , Prague , Czech Republic
5 Prague Zoological Garden , Prague , Czech Republic
Edwards Scott
Electronic publication date: 2019 Feb 6
Publication date: 2019
Volume: 7
Electronic Location ID: e6241
Received 2018 Oct 29; Accepted 2018 Dec 8
Copyright: © 2019 Mazzoleni et al.
Copyright year: 2019
Copyright holder: Mazzoleni et al.
License: This is an open access article distributed under the terms of the Creative Commons Attribution License, which permits unrestricted use, distribution, reproduction and adaptation in any medium and for any purpose provided that it is properly attributed. For attribution, the original author(s), title, publication source (PeerJ) and either DOI or URL of the article must be cited.
License URL: https://creativecommons.org/licenses/by/4.0/

Keywords: Comparative genome hybridization, FISH, Sex determination, Evolution, Telomeres, Microsatellite, Karyotype, Turtles, Sex chromosomes

Funding: Charles University Project PRIMUS/SCI/46 Grant Agency of Charles University GAUK 444217 Charles University Research Centre Program 204069 The project was supported by the Charles University Project PRIMUS/SCI/46, by the Grant Agency of Charles University (GAUK 444217), and Charles University Research Centre Program (204069). The funders had no role in study design, data collection and analysis, decision to publish, or preparation of the manuscript.

==============================
For a long time, turtles of the family Geoemydidae have been considered exceptional because representatives of this family were thought to possess a wide variety of sex determination systems. In the present study, we cytogenetically studied Geoemyda spengleri and G. japonica and re-examined the putative presence of sex chromosomes in Pangshura smithii. Karyotypes were examined by assessing the occurrence of constitutive heterochromatin, by comparative genome hybridization and in situ hybridization with repetitive motifs, which are often accumulated on differentiated sex chromosomes in reptiles. We found similar karyotypes, similar distributions of constitutive heterochromatin and a similar topology of tested repetitive motifs for all three species. We did not detect differentiated sex chromosomes in any of the species. For P. smithii, a ZZ/ZW sex determination system, with differentiated sex chromosomes, was described more than 40 years ago, but this finding has never been re-examined and was cited in all reviews of sex determination in reptiles. Here, we show that the identification of sex chromosomes in the original report was based on the erroneous pairing of chromosomes in the karyogram, causing over decades an error cascade regarding the inferences derived from the putative existence of female heterogamety in geoemydid turtles.

Introduction

Turtles exhibit different sex determination modes. Although it is still a matter of debate, the ancestral (Valenzuela & Adams, 2011; Johnson Pokorná & Kratochvíl, 2016) and most common sex determination mechanism in turtles is most likely environmental sex determination (ESD). Genotypic sex determination (GSD) evolved independently in five families (Chelidae, Emydidae, Geoemydidae, Kinosternidae, Trionychidae) (Valenzuela & Adams, 2011; Badenhorst et al., 2013). Turtles of the family Geoemydidae (Old World pond turtles) are a fascinating model for the evolution of sex determination because it has been reported that this large family with more than 70 species (Rhodin et al., 2017) includes lineages with ESD as well as GSD, with both male (XX/XY sex chromosomes) and female (ZZ/ZW) heterogamety (Valenzuela & Adams, 2011).

Environmental sex determination was reported for three geoemydid genera, namely Mauremys (including Chinemys) (Rhodin et al., 2017), Melanochelys and Rhinoclemmys, mainly based on skewed sex ratios of hatchlings incubated at different temperatures (Ewert, Etchberger & Nelson, 2004). So far, cytogenetic examinations revealed XX/XY sex chromosomes only in the black marsh turtle Siebenrockiella crassicollis (Carr & Bickham, 1986; Kawagoshi, Nishida & Matsuda, 2012) and ZZ/ZW sex chromosomes only in the brown roofed turtle Pangshura smithii (Sharma, Kaur & Nakhasi, 1975). The XX/XY sex chromosomes of S. crassicollis are medium-sized and have been assigned as the fourth pair of the karyogram. The sex chromosomes are heteromorphic and with gene content partially homologous to chromosome pair five of chicken (Gallus gallus) and Chinese softshell turtle (Pelodiscus sinensis) (Kawagoshi, Nishida & Matsuda, 2012). The Y chromosome is metacentric, and the X chromosome is submetacentric, with a prominent C-positive band, missing on the Y. Despite that the X and Y chromosomes differ in morphology and C-banding pattern, it seems that they share gene content extensively. Sex-specific regions were not detected after single-copy gene mapping (Kawagoshi, Nishida & Matsuda, 2012). Therefore, we assume that these sex chromosomes are at an early stage of differentiation with a small sex-specific region. For P. smithii, ZZ/ZW sex chromosomes have been reported by Sharma, Kaur & Nakhasi (1975) based on distinct chromosome morphology. For the majority of species of the family Geoemydidae, the sex determination mode remains unstudied.

In the current investigation, we cytogenetically explored the brown roofed turtle P. smithii, the black-breasted leaf turtle Geoemyda spengleri and the Ryukyu black-breasted leaf turtle G. japonica. The genus Geoemyda is especially interesting because it represents the sister taxon of Siebenrockiella, the genus with evident male heterogametic sex chromosomes (Carr & Bickham, 1986; Kawagoshi, Nishida & Matsuda, 2012). In addition, the genus Geoemyda is phylogenetically nested in a major geoemydid clade containing also Pangshura (Spinks et al., 2004; Lourenço et al., 2013; Pereira et al., 2017), a genus with reported female heterogametic sex chromosomes (Sharma, Kaur & Nakhasi, 1975).

The pioneering studies by Nakamura (1937, 1949) reported a chromosome number of 2n = 52 for G. spengleri, but karyotypes were not documented photographically. In addition, neither the sex of the examined turtles nor their geographical origin was reported. We assume that Nakamura (1937, 1949) actually studied G. japonica, since the Japanese populations had the status of a subspecies of G. spengleri at that time (Yasukawa, Ota & Hikida, 1992). Chaowen, Ming & Liuwang (1998) studied later undoubtedly G. spengleri, using individuals originating from Hunan Province, China. Chaowen, Ming & Liuwang (1998) applied classic cytogenetic methods and revealed also a karyotype of 2n = 52 chromosomes. However, no other cytogenetic approach has been applied to Geoemyda yet. Besides the karyogram of P. smithii published by Sharma, Kaur & Nakhasi (1975), no further cytogenetic studies exist for this species.

In the present study, we constructed karyograms for all three species and further explored their karyotypes by C-banding stain to reveal the distribution of constitutive heterochromatin. Furthermore, we examined the presence of differentiated sex chromosomes by comparative genome hybridization (CGH) and fluorescence in situ hybridization with repetitive elements that often accumulate on sex chromosomes of reptiles, such as telomeric motifs, (GATA)8 microsatellite repeats and rDNA loci (Literman, Badenhorst & Valenzuela, 2014; Matsubara et al., 2016; Rovatsos et al., 2017a; Augstenová et al., 2018).

Materials and Methods

Samples and species verification

Blood samples from four individuals of G. japonica, five individuals of G. spengleri and four individuals of P. smithii (Table 1) were used for preparation of mitotic chromosome suspensions and DNA isolation. All turtles are captive-bred or legally imported, and kept in Zoo Plzeň (Czech Republic), Prague Zoo (Czech Republic), or the Museum of Zoology, Senckenberg Dresden (Germany).

Table 1 Number of individuals per species and sex, analyzed in this study.

Species	Sex	
♂	♀	
Geoemyda japonica	2	2	
Geoemyda spengleri	3	2	
Pangshura smithii	2	2	

Genomic DNA was extracted using a DNeasy Blood and Tissue Kit (Qiagen, Venlo, Netherlands). We amplified by PCR and sequenced the mitochondrial cytochrome b gene (cyt b), in order to verify the taxon and to provide a DNA-based identity of our cytogenetically examined material for future comparison (for the same approach see Koubová et al., 2014; Rovatsos et al., 2015a, 2016a; Rovatsos, Johnson Pokorná & Kratochvíl, 2015b). Cyt b was amplified by PCR using the primers L14919 5′-AACCACGGTTGTTATTCAACT-3′ and H16064 5′-CTTTGGTTTACAAGAACAATGCTTTA-3′ (Burbrink, Lawson & Slowinski, 2000; De Queiroz, Lawson & Lemos-Espinal, 2002). The PCR reaction protocol consists of 20–80 ng of DNA, one μl of each primer (10 pmol/μl), five μl of 10× PCR buffer (Bioline GmbH, Luckenwalde, Germany), 2.5 μl of MgCl2 (50 mM), one μl of dNTPs (10 mM each), 0.5 μl of BioTaq DNA polymerase (5 U/μl, Bioline) and water up to final volume of 50 μl. The amplification conditions were: 95 °C for 3 min, followed by 35 cycles of 95 °C for 30 s, 50 °C for 30 s, and 72 °C for 1 min, and the final step of 72 °C for 5 min. The PCR products were sequenced by Macrogen (Seoul, South Korea), and the obtained sequences deposited in GenBank. A BLAST search (Altschul et al., 1990) was performed to compare our sequences with those previously deposited in public databases.

Chromosome preparation and staining

Mitotic chromosome suspensions were prepared from all studied individuals using whole blood cell cultures. For leukocyte cultivation, 100–300 μl of blood samples were cultured at 30 °C for a week without CO2 supplementation in 5 ml of DMEM medium (Gibco) enriched with 10% fetal bovine serum (Gibco, Carlsbad, CA, USA), 100 μg/ml lipopolysacharide (Sigma-Aldrich, St. Louis, MO, USA), 2 mM L-glutamine (Sigma-Aldrich, St. Louis, MO, USA), 3% phytohaemaglutinin M solution (Gibco), 100 units/ml of penicillin and 100 μg/ml of streptomycin (Gibco). Three hours before harvesting, 35 μl of colchemid solution (10 μg/ml stock solution, Roche, Basel, Switzerland) was added to the medium. Chromosome suspensions were obtained according to the standard method, including an initial hypotonic treatment with 0.075M KCl at 37 °C for 30 min and four times fixation in 3:1 methanol/acetic acid solution. Chromosome suspensions were stored in a freezer for further use.

Chromosomal spreads were stained with Giemsa solution, and selected metaphases were captured in a Provis AX70 (Olympus Corporation, Tokyo, Japan) fluorescence microscope, equipped with a DP30BW digital camera (Olympus). Subsequently, karyograms were constructed using Ikaros karyotyping software (Metasystems, Altlussheim, Germany).

The distribution of constitutive heterochromatin was detected by C-banding (Sumner, 1972). The slides were aged at 55 °C for 1 h, then soaked successively in 0.2N HCl at room temperature for 45 min, in 5% Ba(OH)2 solution at 45 °C for 4–5 min and in 2xSSC for 1 h at 60 °C, with intermediate washes in distilled water, and finally stained with 4′,6-diamidino-2-phenylindole (DAPI) and mounted with antifade medium Vectashield (Vector Laboratories, Burlingame, CA, USA).

Fluorescence in situ hybridization with probes for repetitive elements

The probe to detect the topology of rDNA loci was prepared from a plasmid (pDm r.a51#1) with an 11.5-kb insertion, encoding the 18S and 28S rRNA units of Drosophila melanogaster (Endow, 1982) and labelled with biotin-dUTP using a Nick Translation Kit (Abbott Laboratories, Chicago, IL, USA).

The probe for telomeric motifs (TTAGGG)n was produced and labelled with biotin-dUTP in a single PCR reaction using the primers (TTAGGG)5 and (CCCTAA)5 without a DNA template (Ijdo et al., 1991). The probes for the detection of rDNA loci and telomeric motifs were ethanol-precipitated with sonicated salmon sperm DNA and subsequently resuspended in hybridization buffer (50% formamide/2xSSC) (Rovatsos et al., 2015a, 2017b).

The probe for the GATA microsatellite motif was synthesized by Macrogen (Seoul, South Korea) as (GATA)8 and labelled with biotin. Subsequently, 0.3 μl of (GATA)8 biotin-labelled probe (100 pmol/μl stock solution) was diluted in 10 μl of hybridization buffer (50% formamide, 20xSSC, 10% sodium dodecyl sulphate, 10% dextran sulphate, 1× Denhard’s buffer, pH = 7) per slide.

The preparation of chromosome spreads and probes, the hybridization conditions, the post-hybridization washes, the signal amplification and detection are explained in detail in Rovatsos et al. (2015a). At least 20 metaphases per slide were captured to confirm the fluorescent signal. The pictures were collected in black and white and superimposed with colors. The photos were processed with DP Manager imaging software (Olympus).

Comparative genome hybridization

To detect putative sex-specific chromosome regions, CGH was used according to our standard protocol (Rovatsos et al., 2015a). In each species, equal amounts of male and female genomic DNA (one μg each) were labelled independently with biotin-dUTP and digoxigenin-dUTP, respectively, using a Nick translation kit (Abbott Laboratories, Chicago, IL, USA) and then mixed together. Sonicated salmon sperm DNA was added and ethanol-precipitation was carried out overnight at −20 °C. The labelled DNA was resuspended in hybridization buffer, denatured at 75 °C for 10 min and immediately chilled on ice for 10 min prior to hybridization. The slides with chromosomal material were subsequently treated with RNase A and pepsin, fixed with 1% formaldehyde, dehydrated through an ethanol series, denatured in 70% formamide/2xSSC at 75 °C for 3 min, dehydrated again and air-dried. Hybridization was performed at 37 °C for 2 or 3 days. Post-hybridization washes were performed three times in 50% formamide/2xSSC at 42 °C for 5 min and twice in 2xSSC at room temperature for 5 min. Afterward, the slides were incubated in 100 μl of 4xSSC/5% blocking reagent (Roche, Basel, Switzerland) at 37 °C for 30 min and then with 100 μl of 4xSSC/5% blocking reagent including avidin-FITC (Vector Laboratories) and anti-digoxigenin rhodamine (Roche, Basel, Switzerland) at 37 °C for 30 min. The slides were washed in 4xSSC/0.05% Tween 20, dehydrated, air dried, stained with DAPI, and mounted with Vectashield (Vector Laboratories).

Results

Species verification

The mitochondrial cyt b gene was successfully amplified by PCR and sequenced in all three examined species. A BLAST search (Altschul et al., 1990) of the obtained sequences verified the expected taxonomic identity of the turtles examined here as G. japonica, G. spengleri, and P. smithii. The haplotypes are deposited in GenBank, under the accession numbers MK097237–MK097240.

Karyotype reconstruction and C-banding

Both G. japonica and G. spengleri have a similar karyotype with 2n = 52 chromosomes composed of 12 pairs of macrochromosomes, gradually decreasing in size, and 14 pairs of microchromosomes. Among macrochromosomes, nine pairs are bi-armed and three are acrocentric (pairs 6, 7, and 11) (Fig. 1). C-positive bands were identified in the centromeric regions of almost all chromosomes. A prominent heterochromatic block has been detected in the chromosome pair 12 in metaphases of both sexes in both species (Fig. 1).

Figure 1 Karyograms and C-banded metaphases of Geoemyda japonica (A–D), Geoemyda spengleri (E–H), and Pangshura smithii (I–L).

Please note that microchromosomes are paired according to size for illustration, which does not correspond to actual homology of chromosomes.

In addition, P. smithii has a similar karyotype with 2n = 52, consisting of 12 pairs of bi-armed macrochromosomes and 14 pairs of microchromosomes. C-positive heterochromatin was detected in the centromeric regions of all chromosomes. An extensive accumulation of constitutive heterochromatin was detected in pair 12 in both sexes. This chromosome pair seems to be polymorphic in size in some individuals, but this polymorphism is not linked to sex.

Fluorescence in situ hybridization and comparative genome hybridization

The rDNA loci are located in the terminal position in a pair of microchromosomes in G. japonica and P. smithii, and near the centromere in a pair of microchromosomes in G. spengleri (Fig. 2). The rDNA loci seem to be in all three species linked to the chromosome pair 12, which has the prominent C-positive heterochromatic block.

Figure 2 FISH with rDNA, (GATA)8 and telomeric probes in metaphases of Geoemyda japonica (A–H), Geoemyda spengleri (I–P), and Pangshura smithii (Q–X).

Chromosomes are stained blue with DAPI, and the signal of the probe is pseudocolored in red. In CGH, the male genome is stained with FITC (green color) and the female genome with rhodamine (red color). Genomic regions common for both sexes appear yellow due to the combination of green and red color. Chromosomal regions with similar sequence content in both sexes are visualized in yellow. Arrows indicate the chromosome pair 12, with the prominent C-positive block.

The telomeric repeats (TTAGGG)n showed the expected terminal chromosome topology in all studied individuals (Fig. 2). The (GATA)8 microsatellite motif had a widespread distribution in several pairs of microchromosomes and in the centromeric region of a pair of acrocentric macrochromosomes in both species of the genus Geoemyda but without any sex-specific signal. A weak signal of the (GATA)8 microsatellite motif was detected in the telomeric regions of several chromosomes in P. smithii but without any sex-specific pattern. CGH did not reveal any sex-specific differences in any of the three species (Fig. 2).

Discussion

Our results confirmed that all three studied species have similar karyotypes with 2n = 52 chromosomes, which agrees with former studies (Nakamura, 1949; Sharma, Kaur & Nakhasi, 1975; Killebrew, 1977; Yasukawa, Ota & Hikida, 1992; Chaowen, Ming & Liuwang, 1998). Geoemyda is phylogenetically close to two lineages (Pangshura, Siebenrockiella) (Spinks et al., 2004; Lourenço et al., 2013; Pereira et al., 2017) for which differentiated sex chromosomes have been reported (Carr & Bickham, 1986; Sharma, Kaur & Nakhasi, 1975; Kawagoshi, Nishida & Matsuda, 2012). In several non-avian reptiles, differentiated sex chromosomes are often highly conserved across the phylogenetic spectrum, for example in trionychid turtles (Rovatsos et al., 2017b), lacertids (Rovatsos et al., 2016b), iguanas (Rovatsos et al., 2014), and caenophidian snakes (Rovatsos et al., 2015c). However, our cytogenetic analysis using multiple approaches did not reveal any differentiated sex chromosomes in G. spengleri and G. japonica. Thus, turtles of this genus have either GSD with poorly differentiated sex chromosomes not detectable by our cytogenetic techniques or ESD where sex chromosomes are lacking (following the definition of ESD by Johnson Pokorná & Kratochvíl (2016)).

We did not detect sex chromosomes in P. smithii despite differentiated, highly heteromorphic ZZ/ZW sex chromosomes had been shown by Sharma, Kaur & Nakhasi (1975) in a karyogram of this species based on Giemsa-stained metaphase chromosomes. In this study, the Z chromosome of P. smithii was identified as a small acrocentric chromosome, while the W chromosome was shown as a medium-sized metacentric chromosome. To explain the discrepancies between our results and those of Sharma, Kaur & Nakhasi (1975), we revisited their karyogram (Fig. 3A) and we discovered several potential errors in their assignment of chromosomes to homologue pairs that likely contributed to the mischaracterization of P. smithii as possessing a ZZ/ZW system (Fig. 3B). Namely, the chromosome identified by Sharma, Kaur & Nakhasi (1975) as the Z chromosome is a microchromosome, and we conclude that it can be better reassigned as a homolog of one of the pairs 16–26. Additionally, the metacentric chromosome identified by Sharma, Kaur & Nakhasi (1975) as the W chromosome could be reassigned as a homolog of pair 7, 8, or 9. After simple rearrangement of the original karyogram, no obviously heteromorphic pair of chromosomes is detectable (Fig. 3B), consistent with our own karyotyping of new specimens (Fig. 3C).

Figure 3 The original karyogram of Sharma, Kaur & Nakhasi (1975) (A), their karyogram re-arranged by us (B), and a new karyogram of a female individual from our studied material (C).

Note that the chromosomes misidentified as Z and W in the original study (A) can be autosomal and easily assigned according to size and morphology into the pairs 16–26 and 7–9, respectively, in our new karyogram (C). Numbers in the re-arranged karyogram (B) refer to the original assignment of chromosome pairs by Sharma, Kaur & Nakhasi (1975).

We found variability in size between the homologous chromosomes in the pair 12 of all three examined species of turtles. This pair includes heterochromatic blocks co-localizing with the accumulation of rDNA repeats (Figs. 1 and 2). Heterochromatic blocks are often connected with autosomal polymorphism due to rapid divergence of repeat numbers (Altmanová et al., 2016), and a polymorphism in chromosome morphology including rDNA genes was reported also in ESD species of geoemydid turtles such as Rhinoclemmys pulcherrima (Carr & Bickham, 1986). The polymorphism of the chromosome pair 12 is not linked to sex in G. spengleri, G. japonica, or P. smithii. Thus, there is no evidence that this pair corresponds to sex chromosomes. In any case, the chromosome pair 12 was not identified by Sharma, Kaur & Nakhasi (1975) as sex chromosomes, although it might contribute to the incorrect pairing of chromosomes in their karyotype (Fig. 3).

According to our results, there is no evidence for female heterogamety with differentiated sex chromosomes in geoemydid turtles of the genus Pangshura. Thus, among turtles, female heterogamety is only known in softshell turtles (Trionychidae) (Badenhorst et al., 2013; Rovatsos et al., 2017b). In the family Geoemydidae, the only reliable identification of sex chromosomes refers to the XX/XY sex determination system of S. crassicollis (Carr & Bickham, 1981; Kawagoshi, Nishida & Matsuda, 2012), while other studied species possess either ESD as most other lineages of the family Geoemydidae with known sex determination (Fig. 4) or, perhaps, GSD with poorly differentiated and homomorphic sex chromosomes.

Figure 4 Phylogenetic reconstruction of the sex determination modes in turtles from the family Geoemydidae.

Phylogenetic relationships follow Spinks et al. (2004), Lourenço et al. (2013) and Pereira et al. (2017).

Unfortunately, the erroneous identification of putative sex chromosomes in P. smithii was influential for scientific literature. It impacted studies examining the cytogenetics of turtles (Martinez et al., 2008; Kawagoshi, Nishida & Matsuda, 2012) and comparative phylogenetic reconstructions as well as reviews of sex determination mechanisms, causing a 40-year error cascade regarding the inferred number of sex chromosome turnovers in amniotes and the evolution of sex determination and genome organization (Modi & Crews, 2005; Gamble, 2010; Valenzuela & Adams, 2011; Badenhorst et al., 2013; Johnson Pokorná & Kratochvíl, 2016; Montiel et al., 2017). The error cascade caused by the putative sex chromosomes of P. smithii illustrates how little we still know about sex determination in reptiles and that even traditionally widely accepted reports of sex determination modes can benefit from re-examination with modern molecular cytogenetic methods and broader species sampling.

Conclusions

We found that G. spengleri, G. japonica, and P. smithii share karyotypes with 2n = 52 chromosomes and a similar topology of constitutive heterochromatin and repetitive motifs. We did not detect differentiated sex chromosomes in any of these species. It is particularly notable in P. smithii, where a ZZ/ZW sex determination system with differentiated sex chromosomes was described more than 40 years ago. This information was repeated in subsequent reviews and phylogenetic analyses on sex determination in amniotes and influenced their outcomes and conclusions. We show that the identification of sex chromosomes in the original report was based on the erroneous pairing of chromosomes in their karyogram. We conclude that additional research is needed in order to clarify the true sex determination mode in the three studied turtle species, which might possess either GSD with poorly differentiated sex chromosomes not detectable by our cytogenetic techniques or ESD as most other lineages of the family Geoemydidae with known sex determination (Fig. 4). Future research should include controlled incubation experiments of eggs to examine the influence of temperature in hatchling sex ratios in G. spengleri, G. japonica, and P. smithii, as well as molecular cytogenetic examination of additional geoemydid species, to gain a better understanding of the evolution of sex determination in this group.

Supplemental Information

Supplemental Information 1 Haplotype sequences from cyt b gene from the turtles cytogenetically examined in this study.

Click here for additional data file.

We would like to express our gratitude to Petr Ráb for providing laboratory space and constant support, to Nuria Viñuela Rodriguez and Jana Thomayerová for technical assistance. We thank the staff of Prague Zoo and Zoo Plzeň for providing blood samples. We also thank Hidetoshi Ota for Japanese literature and Christian Schmidt for its translation into English.

Additional Information and Declarations

Competing Interests

Author Contributions

Animal Ethics

Data Availability

The authors declare that they have no competing interests.

Sofia Mazzoleni conceived and designed the experiments, performed the experiments, analyzed the data, contributed reagents/materials/analysis tools, prepared figures and/or tables, authored or reviewed drafts of the paper, approved the final draft.

Barbora Augstenová analyzed the data, prepared figures and/or tables, approved the final draft.

Lorenzo Clemente performed the experiments, analyzed the data, prepared figures and/or tables, approved the final draft.

Markus Auer contributed reagents/materials/analysis tools, approved the final draft.

Uwe Fritz contributed reagents/materials/analysis tools, approved the final draft.

Peter Praschag contributed reagents/materials/analysis tools, approved the final draft.

Tomáš Protiva contributed reagents/materials/analysis tools, approved the final draft.

Petr Velenský contributed reagents/materials/analysis tools, approved the final draft.

Lukáš Kratochvíl conceived and designed the experiments, analyzed the data, contributed reagents/materials/analysis tools, prepared figures and/or tables, authored or reviewed drafts of the paper, approved the final draft.

Michail Rovatsos conceived and designed the experiments, performed the experiments, analyzed the data, contributed reagents/materials/analysis tools, prepared figures and/or tables, authored or reviewed drafts of the paper, approved the final draft.

The following information was supplied relating to ethical approvals (i.e., approving body and any reference numbers):

Blood samples were collected by veterinaries primarily for diagnostic purposes. The animals were not handled by the researchers or accommodated in our faculty animal facilities. According to Czech law, such procedure is not qualified as an experiment on animals and does not require approval of the Ethical Committee. The study was performed by a researcher accredited for making experiments on animals by the Ministry of Agriculture of the Czech Republic (Michail Rovatsos, accreditation CZ-03540).

The following information was supplied regarding data availability:

The raw data is included in the figures, in Genbank (MK097237–MK097240) and in the Supplemental Information.

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
