# Peer review of "Turtles of the genera Geoemyda and Pangshura (Testudines: Geoemydidae) lack differentiated sex chromosomes: the end of a 40-year error cascade for Pangshura"

_PeerJ, doi:10.7717/peerj.6241_

## Round 0.1 · original submission · Minor Revisions

It looks like the reviews are very positive and only a few minor issues that reviewer #2 raised need to be addressed. Please send back the revisions and a rebuttal letter as soon as practical.

Reviewer 1 ·

Basic reporting

The manuscript is clearly written and is to the point. It addresses an important hypothesis in that it corrects a long standing misinterpretation of the sex determining mode in a clade of turtles. The coverage of the literature is adequate to establish the problem addressed and its significance. The paper is succinct and addresses a well defined question.

Experimental design

The authors reinterpret the karyotype of earlier workers and present an alternative interpretation that does not require proposing ZZ/ZW chromosomes in a hard-shelled turtle, particularly where a sister taxon has XX/XY chromosomes or GSD.

The authors then use contemporary techniques of CGH and FISH in an attempt to demonstrate sex specific chromosomal differences in 2 males and 2 females of their target species, and fail to identify any differences. They also examine the distribution across metaphase spreads of two common repeats -- GATA and telomeric repeats. I found this a bit odd, as novel repeats on de novo chromosomes could be based on any motif, and not necessarily one of the two motifs chosen. A more exhaustive examination of potential microsatellite repeats would have been more appropriate.

Overall, though, I found the approach to be appropriate and the results convincing.

Validity of the findings

This paper reports a negative result in the context of previous work that had yielded a positive result. Making such a case is always difficult, but in this case the reinterpretation of earlier work and the application of more contemporary approaches to detecting sex linked chromosomal markers is convincing. The authors do not overspeculate.

Additional comments

This is an important paper because it reverses what is clearly a misinterpretation of the chromosomes of P. smitthii. It will be good to see this work published.

Reviewer 2 ·

Basic reporting

Please see General comments for the author below

Experimental design

Please see General comments for the author below

Validity of the findings

Please see General comments for the author below

Additional comments

I am pleased to receive this manuscript to review again. I was positive the first time I reviewed it, and the authors corrected most of the issues raised in the previous review. There are still some pending issues or new issues that need addressing and which I detail below. Thus my review contains some of text from my previous review updated to bring up the issues that still need resolving.

The present manuscript describes the results from a molecular/classic cytogenetics study of three turtle species from the family Geoemydidae in search of sex chromosomes. Some of the data are first time reports for some of this taxa. The data appear robust and indicate that Geoemyda japonica, Geoemyda spengleri, and Pangshura smithii all lack a sex chromosome system detectable at the resolution of the CGH methods used in this study. Importantly, the new data from Pangshura smithii refute a previous report of ZZ/ZW in this species from 1975 that used only Giemsa staining and what appears to be a erroneous assignment of homologous chromosomes. Thus, the results of this study constitute an important contribution not only to cytogenetics but also to evolutionary biology as they alter our understanding of the evolution of sex determination and sex chromosomes in turtles, and I am enthusiastic in seeing this work published. However, the manuscript suffers from some deficiencies, mainly in presentation and grammar, that need addressing before publication. I detail my concerns below.

Table 1. The provenance of each of the animals examined must be included in the text or table. Were these animals from the wild (and if so from which location), or captive bred (and if so, from a zoo, a pet store, a private collection)?

Ln 77: Please clarify why the authors are certain that Chaowen studied G. spengleri? They do explain nicely why Nakamura studied G. japonica, but the same clarification would be useful for Chaowen’s study.

Ln 112-117: The methods here need more detail. The blood culture and chromosome preparation protocols needs description so that the study can be replicated by the scientific community. The authors should include a brief description even if they cite Pokorna et al 2010 as the source of the protocol as this should be a standalone manuscript. Same for CGH in Ln 144. Also, the other version of this paper that I reviewed stated that the C banding was done with modifications to Sumner’s C banding but those were not described? This version simply omitted the mention of the modification, when the authors should instead include the modifications that were done. This again is crucial for the reproducibility of the study by others.

Ln 150: Replace ‘our study material’ with ‘the specimens examined here’.

Figure 3. Panels a and c should be kept as they are. However, all the reassignments of chromosomes to pairs need to be clearly shown in panel b (not just mention this in the text where the description is harder to follow), but only the ZW chromosomes are traced in panel b). For instance, the authors moved the W from the ZW pair to pair 6, and so you should show where the chromosome from pair6 that is no longer there move to (which seemed to have been assigned to pair 9). Furthermore, a new pair 6 is now assigned in panel b compared to panel a. And so on and so forth. Perhaps panel b would be better as a two column (or two row) figure containing the original and the new arrangement of chromosomes, where the correspondence of chromosomes to pairs is denoted by lines connecting them. Attached is a crude (and incomplete example of such a figure, showing the correspondence for only some of the chromosomes).

Ln 202-204. Odd wording. I suggest replacing “identified as a chromosome from the pairs 16 to 26’ with ‘better reassigned as a homolog of one of the pairs 16 to 26’, or something to that effect.

ln 204-206. Likewise, please reword as: ‘Additionally, the metacentric chromosome identified by Sharma et al. (1975) as the W chromosome should be reassigned as a homolog of pair 7, 8 or 9’.

ln 207. Again, I suggest rewording here. Perhaps ‘After simple rearrangement of the original
karyogram, no obviously heteromorphic pair of chromosomes is detectable (Figure
3b), consistent with our own karyotyping of new specimens (Figure 3c).

Ln 216. Replace ‘and’ with ‘or’.

Ln 218. Add ‘the’ before ‘incorrect’.

Annotated reviews are not available for download in order to protect the identity of reviewers who chose to remain anonymous.

---

## Round 0.2 · accepted · Accept

Great job! Have a great holiday!

#